# Measurement of Enantiomer Ratios for Five Monoterpenes From Six Conifer Species by Cartridge Tube-Based Passive Sampling Adsorption/Thermal Desorption (ps-ATD)

Ying Wang,[1,2] Wentai Luo[2], Todd N. Rosenstiel[3], James F. Pankow[2*]

[1]Key Laboratory of Songliao Aquatic Environment
Ministry of Education, Jilin Jianzhu University
5088 Xincheng Street -Jingyue Economic Development District
Changchun 130118, China

[2]Department of Civil and Environmental Engineering
Portland State University
Portland, OR, 97207, United States

[3]Department of Biology
Portland State University
Portland, OR, 97207, United States

*Corresponding author: pankowj@pdx.edu

**Abstract**

Many monoterpenes have at least two different stereochemical forms, and many biosynthetic pathways have long been known to favor one product over the other(s). A rapid method was developed and used in the determination of the (–/+) enantiomeric distributions for $\alpha$-pinene, $\beta$-pinene, camphene, limonene, and $\beta$-phellandrene as emitted by plant material from six conifer species. The six species included two pine species *Pseudotsuga menziesii* and *Pinus ponderosa*, and four cypress species, *Chamaecyparis lawsoniana*, *Thuja plilcata, Juniperus chinensis, and Thuja occidentalis.* The method involved passive sampling adsorption/thermal desorption (ps-ATD). During sampling, the cartridge tube was placed in a 60 mL glass vial with plant material for 1 h. Sample analytes were thermally transferred to a chiral gas chromatography (GC) column. Detection was by mass spectrometry (MS). The six species exhibited different emission patterns for the five monoterpenes in the –/+ totals, although within a given species the distributions among the five monoterpenes were similar across multiple plants. $\beta$-pinene dominated in *P. menziesii* and *P. ponderosa*, and $\alpha$-pinene dominated in *T. plicata* and *T. occidentalis*. The chiral separations revealed differences in the –/+ enantiomeric distributions among the species. The (–) enantiomers of $\alpha$-pinene and $\beta$-pinene dominated strongly in *P. menziesii* and *P. ponderosa*; the (–) enantiomer of $\beta$-phellandrene dominated in *C. lawsoniana*. The method precision was excellent.

**Key words**: monoterpenes, enantiomers, chiral distributions, conifers, passive sampling, ATD, ps-ATD

**Introduction**

Atmospheric emissions of gaseous non-methane organic compounds from plants are both substantial and chemically complex (Guenther et al., 1995, Pankow *et al.,* 2012; de O. Piva et al., 2019). Plant emissions are greater than those from animals, and are believed to be related to a variety of purposes, including repulsion of herbivorous insects and attraction of pollinators and parasites of herbivores (Dicke and Loon, 2000). Isoprene ($C_5H_8$) and compounds derived from isoprene are particularly prominent in plant emission profiles. Guenther et al. (1995) has estimated that isoprene and monoterpenes constitute approximately 11 and 55%, respectively, of global non-methane emissions. Their oxidation in the atmosphere leads to products that promote formation of ozone (Porter et al., 2017) and which condense as secondary organic aerosol particulate matter (Pankow 1994a; Pankow, 1994b; Zhang et al., 2018).

Monoterpenes that possess chiral carbons can exist in two mirror-image "enantiomeric" forms; for *α*-pinene, (–)-*α*-pinene and (+)-*α*-pinene. For a given compound, different biochemical synthesis pathways in different plants can favor one enantiomer over the other, and many biochemical interaction loci are chiral (López et al., 2011). An example pertains to carvone. The form predominantly found in carraway seeds (*Carum carvi*) is *S*-(+)-carvone while the form predominantly found in spearmint (*Mentha spicata*) is *R*-(–)-carvone.

In forests, where legion species are emitting innumerable compounds for which many have multiple enantiomers, the matter is obviously exceedingly complex. For example, it required careful study by Williams et al. (2007) just to be able to conclude that in tropical forests, emission of (−)-α-pinene is light-dependent, and that in boreal forests emission of (+)-α-pinene

is temperature-dependent. Stephanou (2007) has argued that careful and data driven studies of chirality will be required to fully understand the mechanisms of atmospheric emission of volatile organic compounds by plants. Accordingly, improvements in the requisite analytical methods will be useful.

Table 1 provides a brief summary of the methodologies used to carry out chiral determinations of plant monoterpenes. Analyte collection has occurred using solvents in various ways, and by using sorption of volatilized (gaseous) analytes in air to plant material.

Following collection, analytes are subjected to quantitation of the enantiomer forms using chiral gas chromatography (GC). Use of solvents has disadvantages in this type of work because of the difficulties posed by the large signal from the organic solvent, and by sensitivity problems when the analytes in the extract are not sufficiently concentrated (injecting tens to hundreds of μL of a liquid solvent into a GC is fraught with difficulties).

Sorptive sampling collection of gaseous monoterpenes can be carried out using passive diffusion-limited transfer into the coatings of solid phase microextraction (SPME) fibers, or active gas flow pulled through a cartridge tube holding an adsorptive packing, as in the "adsorption/thermal desorption" (ATD) method. For sampling and placement of analytes on a GC column, SPME can lead to better chromatographic resolution than ATD: less time/gas volume is needed to thermally transfer the analytes from the sorption phase to the column. Automated SPME is more logistically fraught than automated ATD, the latter being well optimized and executable on multiple commercial automated instrument platforms. Since ATD interfaced with chiral GC in our laboratory has been found to give more than adequate

enantiomeric resolution for monoterpenes of interest, the goal of this work was to develop and test "ps-ATD" as a simple and low-labor method for carrying out enantiomeric analyses of monoterpenes emitted by plant materials. The method is based on passive-sampling with ATD cartridges followed by automated ATD. Since only enantiomeric *fractions* and not actual enantiomer *concentrations* were sought in this work, use of passive diffusion sampling carried no drawbacks (diffusion coefficients of enantiomer pairs are identical).

## 2 Materials and Methods

### 2.1 Plant Samples

**Purchased Nursery Plants (Six Species).** Six coniferous species were purchased as ~1 m high potted (~8 L pots) saplings from a local nursery in January of 2018. These included the two pine species *Pseudotsuga menziesii* (4 plants) and *Pinus ponderosa* (3 plants), and the four cypress species *Chamaecyparis lawsoniana*, *Thuja plicata*, *Juniperus chinensis* and *Thuja occidentalis* (4 plants each). The saplings were placed on the roof of the SRTC Building on the PSU campus, and were watered daily. The high/low temperature ranges for Portland during 2018 were: March, 19.4/4.3 ºC; April, 30.0/6.7 ºC; May, 31.7/12.3 ºC; June, 34.4/13.1 ºC; July, 35.6/16.7 ºC; August, 35.0/16.6 ºC. The elevation of the PSU campus is 52 m. A foliage sample was collected from each plant at mid height in March 2018 and again in June/July 2018 using a clean pruning shears. The samples were taken immediately to the laboratory for processing.

**Purchased Nursery *T. occidentalis* – Time of Day Samples.** Foliage samples from the purchased *T. occidentalis* plants were collected at mid height with clean shears on August 20,

2018 at 6 AM, 1 PM, 7 PM, and 9 PM. The temperatures and light intensities were recorded. The samples were taken immediately to the laboratory for analysis.

**Established Residential *T. occidentalis*.** Samples from 6 to 7 established (5+ years), ~3+ m tall) specimens of *T. occidentalis* were collected between February 13-26, 2018 from residential locations in each of three suburban vicinities in Oregon (Hillsboro, Seaside, and Sandy). The approximate time of day for the sampling, the annual mean high/low temperatures, the annual mean precipitation, and the elevation for each were as follows: Hillsboro, 6:30 to 7:30pm, 17.2 °C/6.7 °C , 97.0 cm, 52 m; Seaside, 8:30 to 10:00am, 13.9 °C/6.7 °C , 191.4 cm, 8 m; and Sandy, 2:00 to 3:30pm, 15.6 °C/6.1 °C, 198.9 cm, 299 m. For each sample, a 15 to 20 cm branch of foliage at ~1.5 m above ground was clipped using a clean shears. The cut end of each sample was wrapped with a wet paper towel at the cut. Each sample was stored in an unzipped ziplock bag with the cut end inside of the bag. The samples from Hillsboro arrived within 14 h and were analyzed immediately. The samples from Seaside and Sandy arrived at the laboratory within 2 h and were processed immediately.

## 2.2 Sample Preparation

Samples were rinsed with deionized water; surface water was removed by blotting with a clean paper towel. Sample material was cut into ~1 cm pieces with a clean laboratory scissors. Subsamples of ~0.3 g were transferred to clear 60 mL "VOA" vials (Restek Corporation, Bellefonte, PA). Each vial was sealed with a 0.125 in. thick PTFE lined septum (Restek Corporation, Bellefonte, PA) and held at 20±0.5 °C for 60 min. Passive sampling with an ATD cartridge then GC/MS analysis proceeded as described below.

### 2.3 Chemical Standards

The five monoterpenes examined here were *α*-pinene, *β*-pinene, camphene, limonene, and *β*-phellandrene. Authentic chiral and racemic standards were purchased from Sigma Aldrich Inc. (St. Louis, MO) at ≥98% purity.

### 2.4 Gas Chromatography (GC)

Relative total amounts of the monoterpenes (total (+/–) *α*-pinene, total (+/–) *β*-pinene, etc.) and the enantiomeric fractions for the (–) forms were determined by GC. The elution order was established by analysis of standards. The chiral column stationary phase was Supelco Beta DEX™ 120 (Supelco Inc., Bellefonte, PA) with 0.25 µm film thickness, 0.25 mm i.d., and 30 m length. After gaseous introduction of each sample into the column, the GC oven temperature program was: 1) hold at 60 °C for 2 min; 2) ramp to 90 °C at 1 °C/min; 3) ramp to 105 °C at 3 °C/min; 4) ramp to 220 °C at 10 °C/min; then 5) hold at 220 °C for 2 min. The gas flow rate through the column was approximately 1.0 mL/min. Figure 1 provides an example of a chromatogram for a sample.

### 2.5 Headspace Sampling, Analyte Transfer to GC, and Mass Spectrometric (MS) Analysis

The "VOA" vials used were from Restek Corporation (Bellefonte, PA). The 40 mL standard vials contained ~1 mg of neat liquid standard. As noted below, the 60 mL vials were loaded with ~0.3 g of plant material. In all cases, sampling proceeded in a passive manner by exposing the inlet end of an ATD gas sampling cartridge to the vial headspace. Before exposure, each cartridge was otherwise wrapped with clean aluminum foil. For standards, sampling of the gas phase involved a 2 s exposure with the cartridge held in the inlet in the headspace of an open vial. For samples, each cartridge was placed in its vial for 2 h with the

vial capped. No flow through into the cartridge was required to acquire adequate analyte mass for any given analysis (~0.05 ng of an enantiomer on an ATD cartridge (or ~0.01 ng on-column) was required to obtain a signal to noise (S/N) ratio of 50:1). Passive sampling was used because the primary interest was the enantiomeric percentages of the subject compounds, and not emission rates or consequent ecosystem concentrations. The ATD cartridges were from Camsco Inc. (Houston, TX), as packed with 100 mg of 35/60 mesh Tenax TA on the inlet side followed by 200 mg of 60/80 mesh Carbograph 1 TD.

ATD cartridges were auto-processed using a TurboMatrix 650 ATD (PerkinElmer Inc., Waltham, MA) unit interfaced to a Leco Pegasus 4D GC×GC-TOFMS (Leco Corporation, St. Joseph, MI) used in 1-D GC mode (*i.e.*, without application of a secondary column). (TOFMS = time of flight mass spectrometer.) In the Turbomatrix 650 unit, the analytes on each ATD cartridge were thermally desorbed (270 °C, 10 min, 40 mL/min He, backflush mode (outlet to inlet) direction) onto an intermediate Tenax-TA focusing trap held at −10 °C. 25 mL/min of the 40 mL/min desorption flow was discarded as "split" flow. The focusing trap was then thermally desorbed at 280 °C for 5 min at 16 psi constant He pressure. About 2 mL/min of the flow passed onto the GC column in the TOFMS unit via a 225 °C transfer line; the remaining ~20 mL/min split flow was discarded. MS data acquisition began upon initiating thermal desorption of the focusing trap.

For $\alpha$-pinene, camphene, limonene and $\beta$-phellandrene, for the percent enantiomer determinations, the MS quantitation ion used was $m/z = 93$. For $\beta$-pinene, $m/z = 69$ was used. For each compound in a given sample, the percent of each enantiomer was calculated using the

area for each deconvoluted peak (in any case of co-elution) for the enantiomer quantitation ion divided by the corresponding sum for both enantiomers. Note here that both enantiomers in a given pair during will have exhibited the exact same: 1) diffusion coefficient during sampling; 2) transfer efficiencies during analysis; and 3) detector sensitivities.

The fractional mass distribution among the five monoterpenes was calculated for each sample using the peak pair sums, each of which was normalized using total ion chromatogram (TIC)-based relative response factors relative to $\alpha$-pinene ($RRF_{\alpha\text{-pinene}}$). Obtained from analyses of replicate ATD cartridges onto which known amounts (~10 ng) of each of monoterpene in 4 μL of methanol/acetone had been loaded (by syringe), the measured TIC $RRF_{\alpha\text{-pinene}}$ values were $\alpha$-pinene, 1:00; $\beta$-pinene, 0.83; camphene, 0.93; limonene, 0.83; and $\beta$-phellandrene, 0.44. Inherent in these calculations of the fractional mass distributions among the five monoterpenes are the assumptions that: 1) the passive sampling rate by gaseous diffusion was the same for all of the compounds; and 2) the desorption transfer efficiencies to the analytical unit were similar for all of the compounds. The first assumption is excellent given their common molecular weight; the second assumption is considered excellent, though unverified for the exact conditions used here.

The average of the above five TIC $RRF_{\alpha\text{-pinene}}$ values (0.81) was used to obtain an estimate of the mass percentage for each sampling of the sum of the five monoterpenes (10 enantiomers) relative to all detected monoterpenes ($=(\Sigma^5/\Sigma^{all})\times100\%$). The LECO software was used to deconvolute: 1) each of the 10 enantiomer TIC peaks for the five compounds; and 2) each of the other compound TIC peaks identified (based on mass spectral matching and GC

retention time window) as probable monoterpenes. The most abundant of these were sabinene and myrcene. The deconvoluted TIC peak areas ($A$) were integrated then used with the TIC response factors with

$$\sum^{5} = \frac{A_{\alpha\text{-pinene}}}{\mathrm{RRF}_{\alpha\text{-pinene}}} + \frac{A_{\beta\text{-pinene}}}{\mathrm{RRF}_{\beta\text{-pinene}}} + \frac{A_{\text{camphene}}}{\mathrm{RRF}_{\text{camphene}}} + \frac{A_{\text{limonene}}}{\mathrm{RRF}_{\text{limoene}}} + \frac{A_{\beta\text{-phellandrene}}}{\mathrm{RRF}_{\beta\text{-phellandrene}}} \quad (1)$$

$$\sum^{\text{all}} = \sum^{5} + \sum_{i}^{\text{other}} \left( \frac{A_{\text{other}}}{0.81} \right)_{i} \quad (2)$$

### 2.6 Statistical Analyses

One-way ANOVA was used to analyze variables such as proportion of monoterpenes and enantiomeric ratios among six species, as well as enantiomeric ratios in *T. occidentalis* under different conditions. Multiple comparisons among different species, different sampling time and different positions were detected using the least significant difference (LSD) test, with a critical significance level of $p$ = 0.05. All analyses were performed using SPSS statistical software (version 27.0, IBM Inc., Armonk, NY, USA).

## 3 Results and Discussion

### 3.1 Proportion of Monoterpenes Among Different Nursery-Purchased Species

Mass percent values among the five target monoterpenes for the six nursery-purchased species and their $(\Sigma^{5}/\Sigma^{\text{all}})\times 100\%$ values are given in Figures 2.a and 2.b. (and Tables 2.a and 2.b). These values were obtained using the combined (enantiomer pair) deconvoluted TIC peak area data for each monoterpene together with the corresponding $\mathrm{RRF}_{\alpha\text{-pinene}}$ values. $\alpha$-pinene and $\beta$-pinene were found to be the dominant monoterpenes in the two pine species *P. menziesii and P. ponderosa*, and $\alpha$-pinene and limonene dominated in *C. lawsoniana*. Limonene

represented more than 90% of the five compounds for *J. chinensis*.

### 3.2 Enantiomer Percentages among Different Nursery-Purchased Species

The percentages of the (-) form for the five compounds in the six species for March and June/July are given in Figures 3.a and 3.b (and Tables 3.a and 3.b). For all species, the results were similar for the two sampling times. The results for the two pine species (*P. menziesii* and *P. ponderosa*) were similar, but the results varied among the four cypress species (*C. lawsoniana*, *T. plicata*, *J. chinensis*, and). In the two pine species, the percentages of the (-) form were >90%, >90%, and >50% for *α*-pinene, *β*-pinene and limonene, respectively. The lowest percentages of the (-) form for *α*-pinene and limonene were observed in *C. lawsoniana* and *J. chinensis*. The lowest percentages of the (-) form for *β*-pinene were observed in *C. lawsoniana* and *T. plicata*. The (-) form of camphene strongly dominated in *C. lawsoniana*. The (-) form of *β*-phellandrene was highest in *C. lawsoniana*.

### 3.3 Enantiomer Percentages in Nursery-Purchased *T. occidentalis* from 6 AM to 9 AM

The percentages of the (-) form for the five compounds in the nursery-purchased *T. occidentalis* plants in one day in August 2018 are given in Figure 4 (and Table 4). The enantiomeric profiles were very similar for the four different sampling times.

### 3.4 Enantiomer Percentages in Nursery-Purchased vs. Residential *T. occidentalis*

The percentages of the (-) form for the five compounds in nursery-purchased and residential *T. occidentalis* plants (sampled in March 2018 and February 2018, respectively) are given in Figure 5 (and Table 5). The enantiomeric profiles were all remarkably similar.

### 3.5 Enantiomer Percentage Method Precision

There are two compounded sources of the estimated standard deviation values $s$ in the percent (–) enantiomer values in Tables 3-5: 1) the compound-dependent plant material standard deviation ($s_{plant}$) and 2) the analytical method variability ($s_{method}$). Since both sources contribute to the standard deviation values in these tables, each $s$ value given is an upper estimate of $s_{method}$ alone. The $s_{method}$ values are, however, dependent on the percent (–) enantiomer value, being driven to zero at percent (–) values of both 0 and 100: if one enantiomer is completely absent, even poor chiral separation and quantitation will lead to exactly 0 and 100. This assumes no contamination by co-eluting compounds, which would also be one source of dependence of $s_{method}$ on the on-column amounts of the enantiomer pair. Assuming that the latter effect is minor, and examining only the $s$ values in the three tables for which the percent (–) enantiomer values fall in the range 30 to 70%, the $s$ value ranges are: Table 3, 0.9 to 14.9; Table 4, 0.1 to 6.1; and Table 5, 1.7 to 5.1. Given the compound nature of these $s$ values, the smallness of the lower limits of these ranges, and the modest nature of the upper limits of these ranges, we conclude that the method here can provide accurate and precise determination of chiral distributions of gaseous monoterpenes.

**Acknowledgements**

This work was financed by in part by the Maseeh Foundation.

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

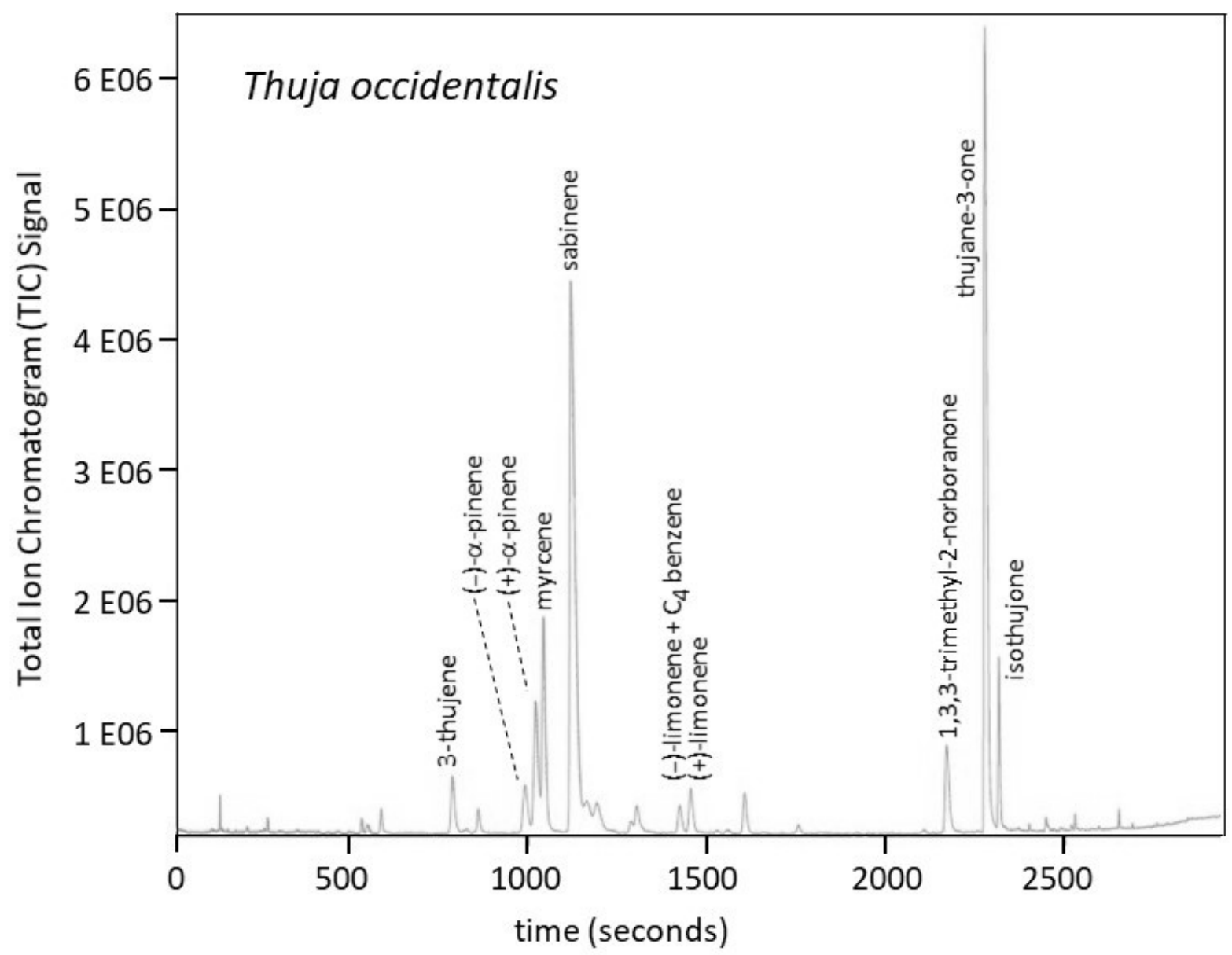

Figure 1. Total ion chromatogram (TIC) by GC/MS (gas chromatography/mass spectrometry) using a Supelco Beta DEX™ 120 chiral capillary column (0.25 µm film thickness, 0.25 mm i.d., and 30 m long; Supelco Inc., Bellefonte, PA) for a *T. occidentalis* sample. The peak marked for (–)-limone contains a contribution from an unidentified $C_4$-benzene. The two α-pinene enantiomers and the two limonene enantiomers were quantitated using the ion $m/z = 93$.

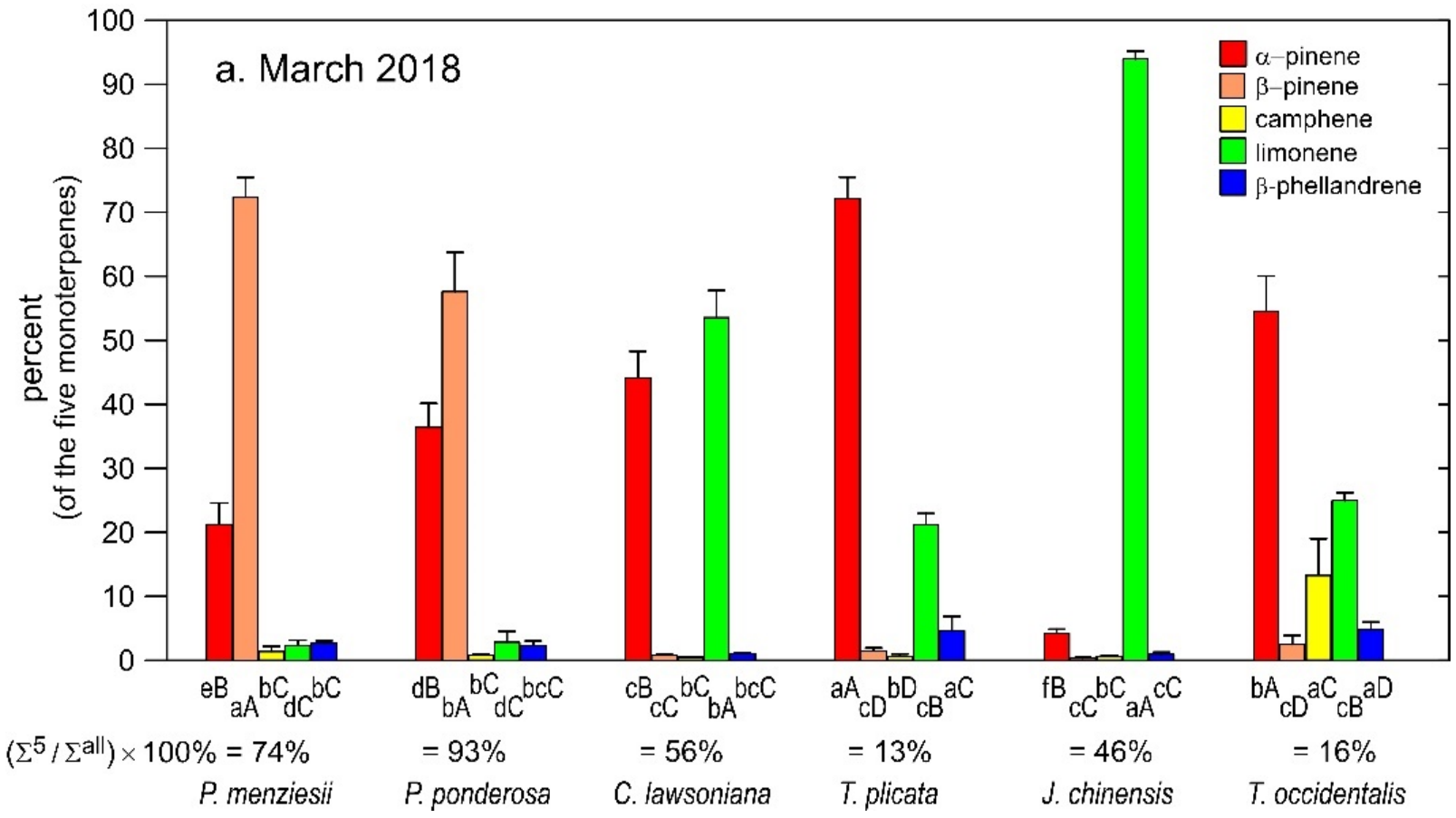

Figure 2.a. Bar graph showing percentages among five monoterpenes in March 2018 for six nursery-purchased conifer species. Within a given species, the same capital letter indicates no significant difference between the monoterpenes. For a given monoterpene, the same lower case letters indicate no significant difference between the species. The percentage values that the five monoterpenes represent as a sum relative to the sum of all detected monoterpenes (=($\Sigma^5/\Sigma^{all}$)×100%) are given. The data values are given in Table 2.a.

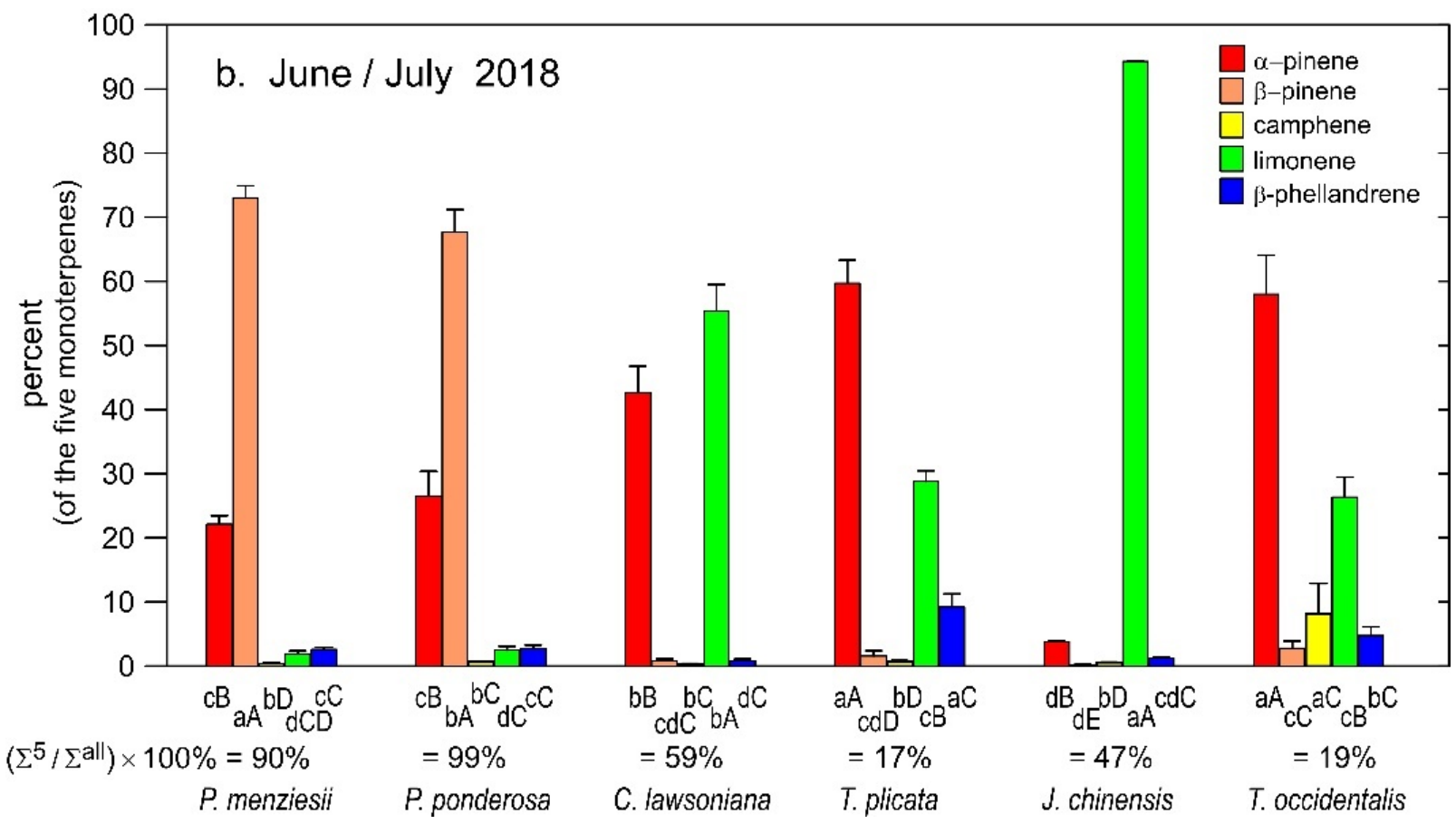

Figure 2.b. Bar graph showing the percentages among five monoterpenes in June/July 2018 for six nursery-purchased conifer species. Within a given species, the same capital letter indicates no significant difference between the monoterpenes. For a given monoterpene, the same lower case letters indicate no significant difference between the species. The percentage values that the five monoterpenes represent as a sum relative to the sum of all detected monoterpenes (=($\Sigma^5/\Sigma^{all}$)×100%) are given. The data values are given in Table 2.b.

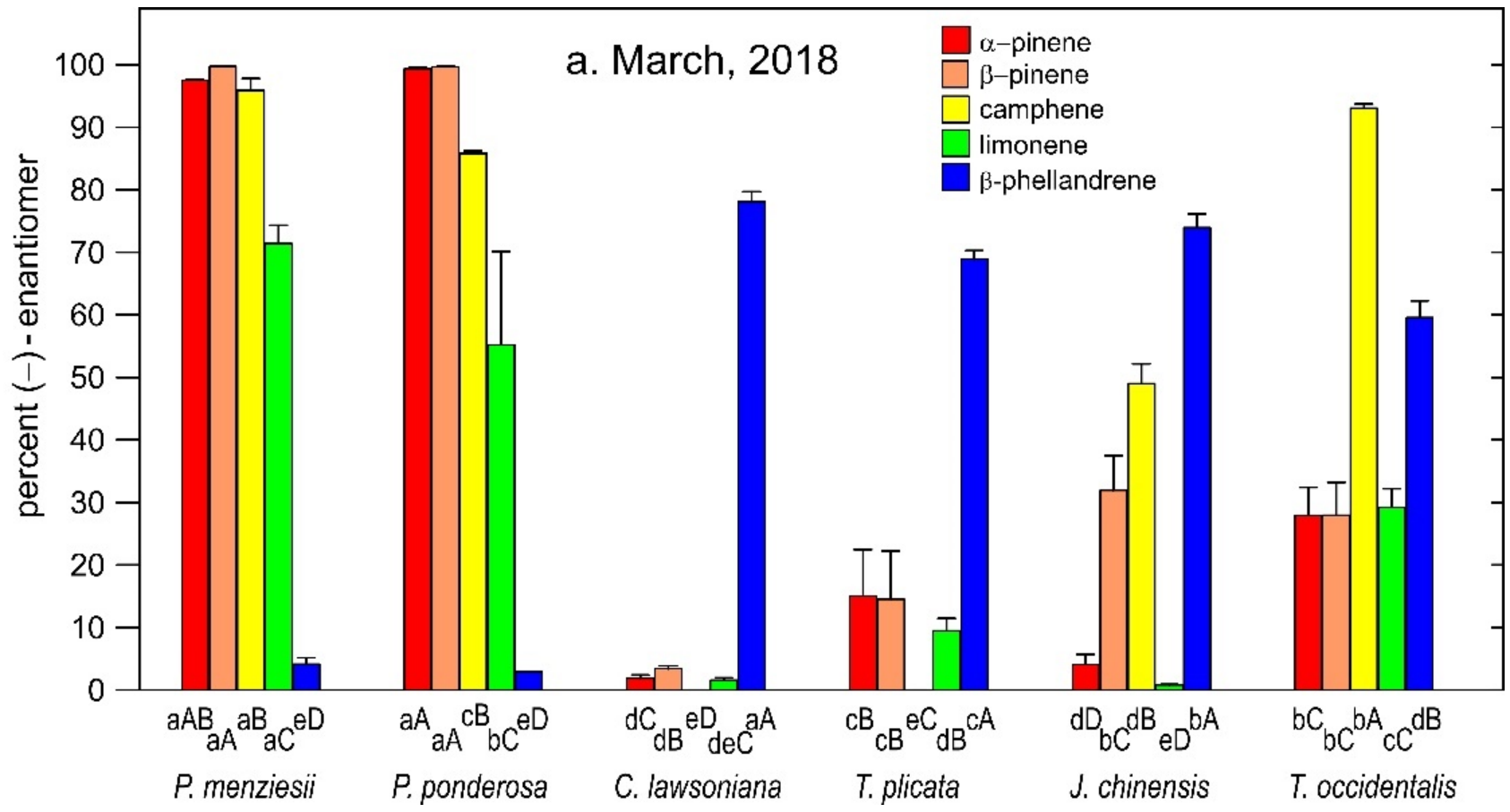

Figure 3.a. Bar graph showing the percentage values for the (-) enantiomer for five monoterpenes in March 2018 for six nursery-purchased conifer species. Within a given species, the same capital letter indicates no significant difference between the monoterpenes. For a given monoterpene, the same lower case letters indicate no significant difference between the species. The data values are given in Table 3.a.

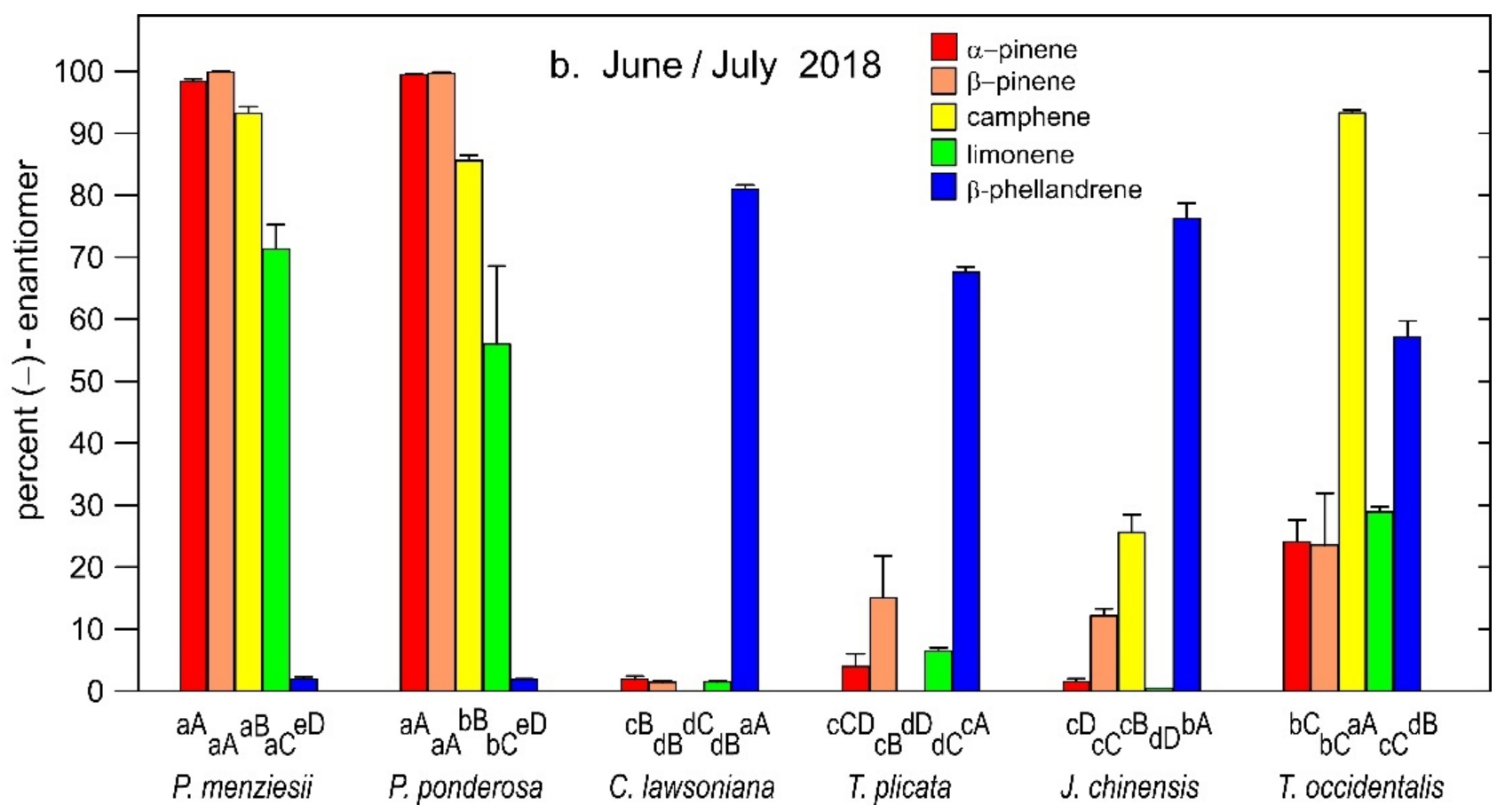

Figure 3.b. Percent of the (-) enantiomer for five monoterpenes in June/July 2018 for six nursery-purchased conifer species. Within a given species, the same capital letter indicates no significant difference between the monoterpenes. For a given monoterpene, the same lower case letters indicate no significant difference between the species. The data values are given in Table 3.b.

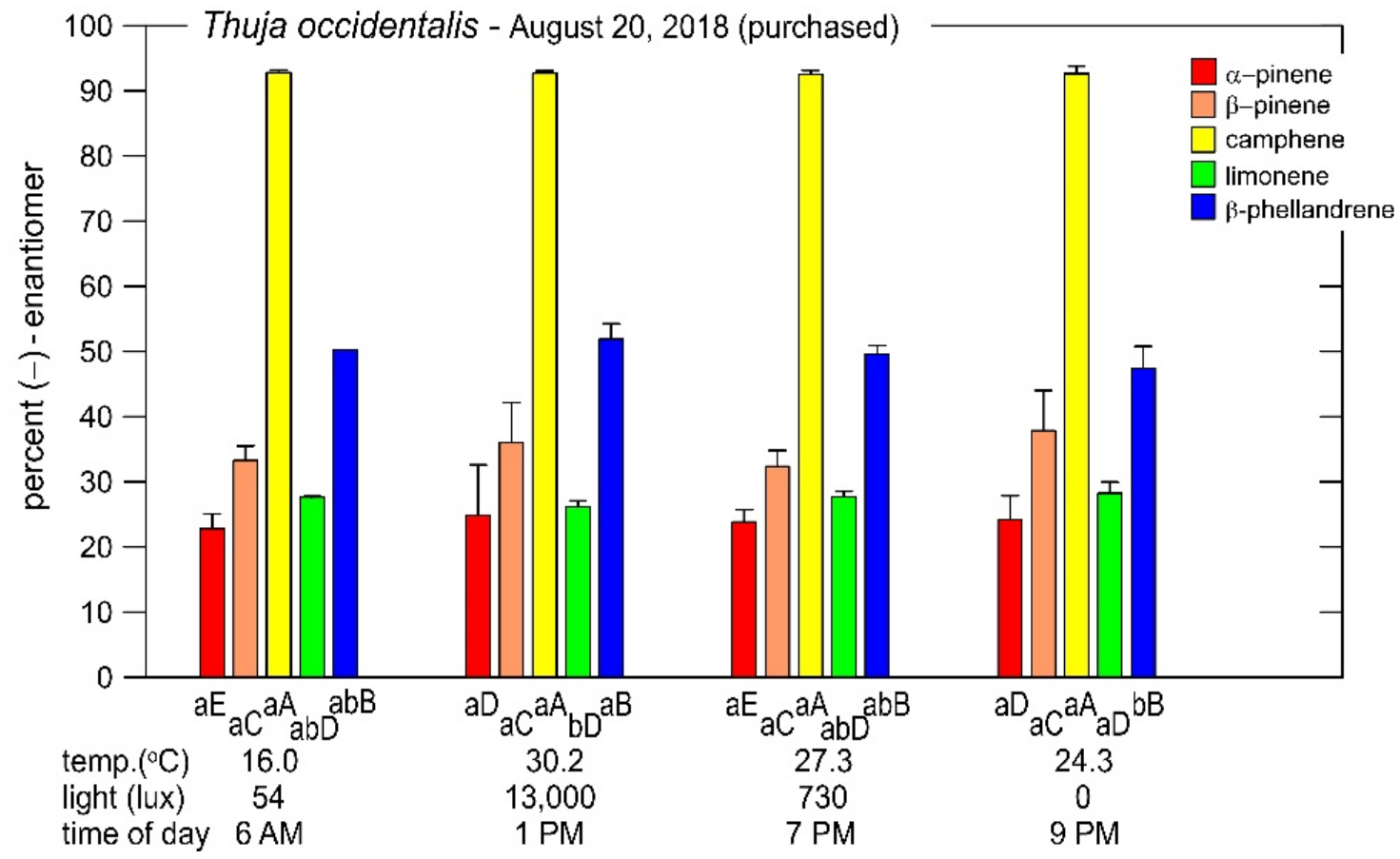

Figure 4. Percent of the (-) enantiomer for five monoterpenes in nursery-purchased samples of *Thuja occidentalis* on August 20, 2018. For a given time, the same capital letter indicates no significant difference between the monoterpenes. For a given monoterpene, the same lower case letters indicate no significant difference between the times. The data values are given in Table 4.

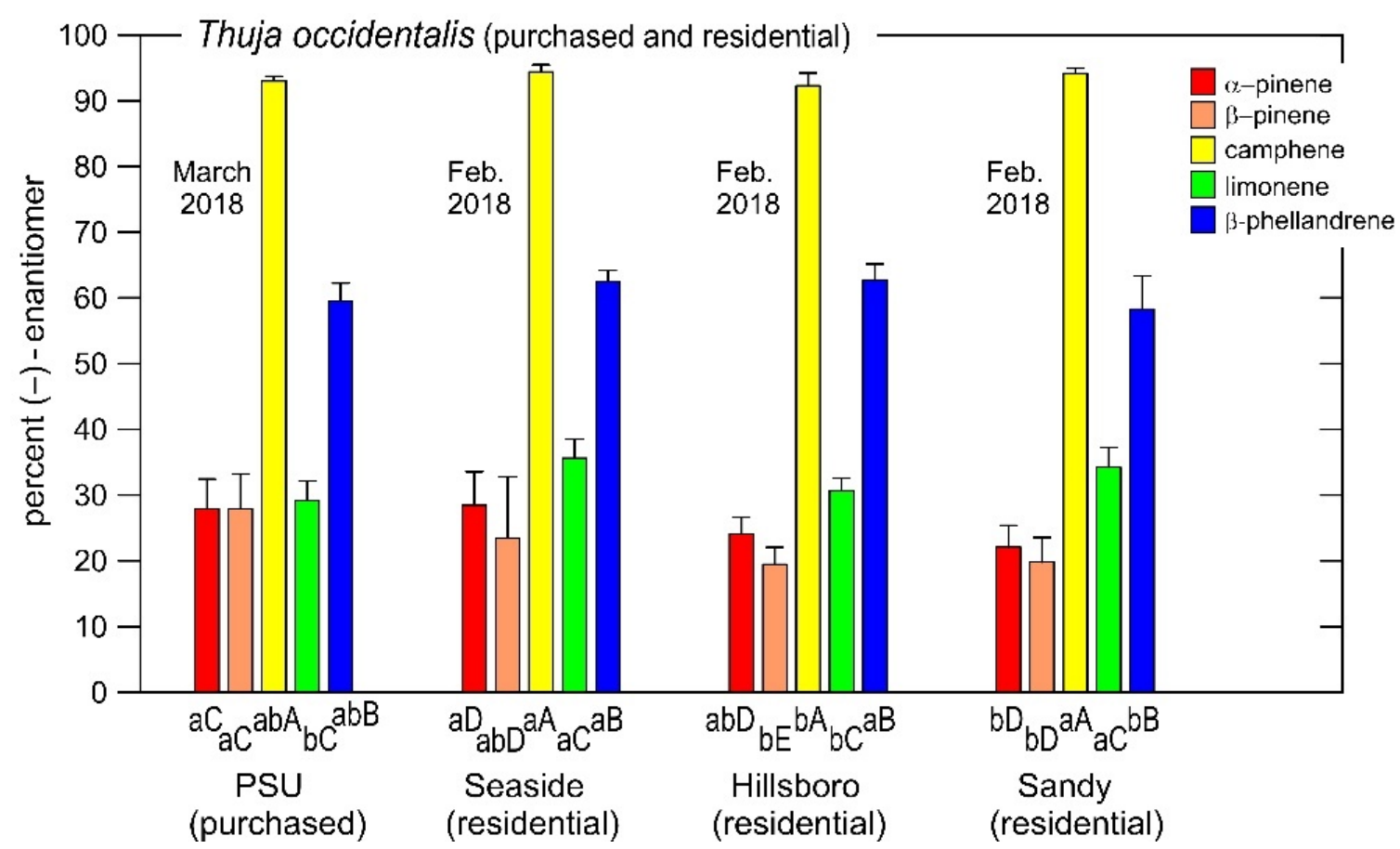

Figure 5. Percent of the (-) enantiomer for five monoterpenes in nursery-purchased (March 2018) and residential (February 2018) samples of *Thuja occidentalis*. For a given sample location, the same capital letter indicates no significant difference between the monoterpenes. For a given monoterpene, the same lower case letters indicate no significant difference between the locations. The data values are given in Table 5. The data for the "PSU (purchased)" plants also appear in Figure 3.a.

**Table 1.** Summary of methods used to obtain then analyze plant-derived chiral biogenic volatile organic compounds.

***Using Solvent and Solvent Injection***

| Citation - Plant/System(s) | Summary |
|---|---|
| Persson *et al*., 1993 *Picea abies* | Method. Hexane extraction of plant material, silica gel clean-up, then two-dimensional heart-cut GC-FID (GC phases: DB-WAX then permethylated *β*-cyclodextrin). Analytes: α-pinene, camphene, *β*-pinene, sabinene, limonene, *β*-phellandrene. |
| Persson *et al.* (1996) *Picea abies* | Method: Hexane extraction of plant material, silica gel clean-up, then two-dimensional heart-cut GC-FID (GC phases: DB-WAX then permethylated *β*-cyclodextrin) for most chiral separations. For 3-carene, a dipentylbutyryl-γ-cyclodextrin phase was used; the constituents of the monoterpenes were identified by mass spectroscopy (MS). Analytes: *α*-pinene, camphene, *β*-pinene, sabinene, limonene, *β*-phellandrene, and others (23 total enantiomers). |
| Sjödin *et al.* (1996) *Pinus sylvestris* | Method: Same as in Persson et al. (1996). Analytes: *α*-pinene, camphene, *β*-pinene, sabinene, limonene, *β*-phellandrene, myrcene, 3-carene. |
| Wibe *et al.* (1998) *Picea abies*, *Pinus sylvestris*, *Juniperus communis* | Method: Sample headspace volatiles with air flow through an adsorbent (Porapak Q), recover analytes with solvent, then two-dimensional heart-cut GC/FID (GC phases: DB-WAX then two permethylated β-cyclodextrin). Analytes: α-pinene, camphene, *β*-pinene, sabinene, 3-carene, limonene, *β*-phellandrene. |
| Ložienė and Labokas (2012) *Juniperus communis L* | Method: Steam distillation collection of essential oils, then dilution in a solvent mix (diethyl ether/pentane), then GC/FID (GC phase: HP-Chiral-20B). Analyte: *α*-pinene. |
| Southwell *et al.* (2017) *Melaleuca alternifolia and M. linariifolia* | Method: Steam distillation collection of essential oils, then dilution with ethanol, then GC/FID (GC phase: cyclodextrin). Analytes: Terpinen-4-ol, limonene, *α*-terpineol. |
| Inoue *et al.* (2018) *Lindera umbellata var. membranacea* | Method: Hexane extraction of plant material, then GC/MS analysis (GC phase: CycloSil-B). Analytes: *α*-pinene, camphene, *β*-pinene, sabinene, limonene, *β*-phellandrene, and others (29 total, including enantiomeric variations). |

***Using Diffusion Sampling by Exposure of SPME Fiber to Air Containing Plant Emissions then Thermal Desorption***

| Citation - Plant/System(s) | Summary |
|---|---|
| Ruiz del Castillo *et al.* (2004) *Mentha piperita* | Method: SPME with 100 µm polydimethylsiloxane (PDMS) phase, then GC/MS (GC phase: permethylated *β*-cyclodextrin or 2,3-di-acetoxy-6-*O*-*tert*-butyl dimethylsilyl γ-cyclodextrin). Analytes: *α*-pinene, camphene, *β*-pinene, sabinene, limonene, *β*-phellandrene, and others (19 total, including enantiomeric variations). |
| Yassaa and Williams (2007) *P. sylvestris chemotype A and B* (boreal coniferous forest) | Method: SPME with PDMS/DVB phase, then GC/MS (GC phase: permethylated *β*-cyclodextrin). Analytes: α-pinene, camphene, *β*-pinene, sabinene, limonene, *β*-phellandrene, and others (17 total, including enantiomeric variations). |

| | |
|---|---|
| Yassaa *et al.* (2010)<br>*Quercus ilex* | Method: SPME with PDMS/DVB phase, then GC/MS (GC phase: $\beta$-cyclodextrin).<br>Analytes: $\alpha$-pinene, camphene, $\beta$-pinene, sabinene, limonene, myrcene, 3-carene, 1,8-cineol, cis-$\beta$-ocimene. |
| ***Using Active Flow Sampling of Air Containing Plant Emissions Through an ATD Sorbent Cartridge Tube then Thermal Desorption*** | |
| Citation - Plant/System(s) | Summary |
| Williams *et al.* (2007)<br>tropical and boreal forests | Method: ATD with Carbograph I/Carbograph II adsorbent, then GC/MS (GC phase: $\beta$-cyclodextrin).<br>Analytes: $\alpha$-pinene, camphene, $\beta$-pinene, limonene, myrcene, 3-carene. |
| Song *et al.* (2011)<br>*Pinus pinea L.* (forest canopy) | Method: ATD with Tenax TA/Carbograph I, then GC/MS (GC phase: $\beta$-cyclodextrin).<br>Analytes: $\alpha$-pinene, $\beta$-pinene, limonene, camphor, and others (12 total including enantiomeric variations). |
| Song *et al.* (2014)<br>*Quercus ilex L., Rosmarinus officinalis L., and Pinus halepensis Mill.* | Method: ATD with Carbograph I/II or Tenax/carbograph, then GC/MS (GC phase: $\beta$-cyclodextrin).<br>Analytes: $\alpha$-pinene, $\beta$-pinene, limonene, camphor, isoprene, and others (13 total including enantiomers). |
| Staudt *et al.* (2019)<br>Maritime pine (forest canopy) | Method: ATD with Tenax TA/Carbograph 1 adsorbent, then GC/MS (GC phase: dimethyl TBS $\beta$-cyclodextrin).<br>Analytes: $\alpha$-pinene, $\beta$-pinene. |
| Zannoni *et al.* (2020)<br>Amazon rain forest | Method: ATD with Carbographs 1 and 5, then GC/MS (GC phase: dimethyl TBS $\beta$-cyclodextrin).<br>Analyte: $\alpha$-pinene. |
| **Using *Passive Diffusion Sampling of Air Containing Plant Emissions Into Open End of ATD Sorbent Tube the Thermal Desorption*** | |
| Citation - Plant/System(s) | Summary |
| This Work<br>*Pseudotsuga menziesii, Pinus ponderosa, Chamaecyparis lawsoniana, Thuja plilcata, Juniperus chinensis, Thuja occidentalis* | Method: ATD with Tenas TA/Carbographs 1 adsorbent, then GC/MS (GC phase: $\beta$-cyclodextrin).<br>Analytes: $\alpha$-pinene, camphene, $\beta$-pinene, limonene, $\beta$-phellandrene. |

**Table 2.** Mass fraction values (including both enantiomers) for each of five chiral monoterpenes over those five monoterpenes, and average values of $(\Sigma^5 / \Sigma^{all}) \times 100\%$ (= mass fractions for the mass sum for those five terpenes over all detected monoterpenes). The nursery-purchased plants were located at PSU and sampled in March 2018 and again in June/July 2018. Number of replicates $N = 4$ for all species, except $N = 3$ for *P. ponderosa*. For each replicate, a separate sample of plant material was analyzed once.

**Table 2.a.** March 2018 (data are plotted in Figure 2.a).

| | mass fractions of five monoterpenes over those five monoterpenes (total = 100%) | | | | | |
|---|---|---|---|---|---|---|
| species | $\alpha$-pinene | $\beta$-pinene | camphene | limonene | $\beta$-phellandrene | $(\Sigma^5 / \Sigma^{all}) \times 100\%$ |
| *P. menziesii* | 21.2 ± 3.3 | 72.4 ± 3.1 | 1.4 ± 0.8 | 2.3 ± 0.8 | 2.7 ± 0.4 | 74 |
| *P. ponderosa* | 36.4 ± 3.8 | 57.6 ± 6.1 | 0.80 ± 0.22 | 2.8 ± 1.7 | 2.4 ± 0.7 | 93 |
| *C. lawsoniana* | 44.1 ± 4.1 | 0.78 ± 0.1 | 0.50 ± 0.10 | 53.5 ± 4.2 | 1.0 ± 0.1 | 56 |
| *T. plicata* | 72.2 ± 3.3 | 1.4 ± 0.5 | 0.59 ± 0.37 | 21.2 ± 1.7 | 4.6 ± 2.2 | 13 |
| *J. chinesis* | 4.2 ± 0.7 | 0.30 ± 0.15 | 0.59 ± 0.17 | 93.9 ± 1.2 | 1.0 ± 0.3 | 46 |
| *T. occidentalis* | 54.5 ± 5.6 | 2.5 ± 1.4 | 13.3 ± 5.7 | 25.0 ± 1.2 | 4.8 ± 1.2 | 16 |

**Table 2.b.** June/July 2018 (data are plotted in Figure 2.b).

| | mass fractions of five monoterpenes over those five monoterpenes (total = 100%) | | | | | |
|---|---|---|---|---|---|---|
| species | $\alpha$-pinene | $\beta$-pinene | camphene | limonene | $\beta$-phellandrene | $(\Sigma^5 / \Sigma^{all}) \times 100\%$ |
| *P. menziesii* | 22.1 ±1.3 | 73.0 ± 1.9 | 0.38 ± 0.15 | 1.9 ± 0.5 | 2.6 ± 0.3 | 90 |
| *P. ponderosa* | 26.5 ± 3.9 | 67.7 ± 3.5 | 0.71 ± 0.11 | 2.5 ± 0.6 | 2.7 ± 0.6 | 99 |
| *C. lawsoniana* | 42.6 ± 4.2 | 0.83 ± 0.31 | 0.33 ± 0.09 | 55.4 ± 4.0 | 0.82 ± 0.27 | 59 |
| *T. plicata* | 59.7 ± 3.6 | 1.6 ± 0.8 | 0.72 ± 0.15 | 28.8 ± 1.7 | 9.2 ± 2.1 | 17 |
| *J. chinesis* | 3.8 ± 0.15 | 0.13 ± 0.15 | 0.54 ± 0.10 | 94.3 ± 0.09 | 1.2 ± 0.2 | 47 |
| *T. occidentalis* | 58.0 ± 6.1 | 2.8 ± 1.1 | 8.1 ± 4.8 | 26.4 ± 3.1 | 4.7 ± 1.4 | 19 |

**Table 3.** Percent (–) enantiomer values ± 1 standard deviation (*s*) for five chiral monoterpenes in six conifer species in nursery-purchased plants located at PSU and sampled in March 2018 and again in June/July 2018. (The data were obtained from the same set of analyses carried out to generate the data in Table 2.)

**Table 3.a.** March 2018 (data are plotted in Figure 3.a).

| species | α-pinene | β-pinene | camphene | limonene | β-phellandrene |
|---|---|---|---|---|---|
| *P. menziesii* | 97.5 ± 0.09 | 99.7 ± 0.1 | 95.9 ± 1.9 | 71.4 ± 2.9 | 4.2 ± 1.0 |
| *P. ponderosa* | 99.3 ± 0.2 | 99.6 ± 0.1 | 85.8 ± 0.5 | 55.2 ± 14.9 | 2.9 ± 0.08 |
| *C. lawsoniana* | 1.9 ± 0.5 | 3.4 ± 0.5 | 0.0 ± 0.0 | 1.6 ± 0.4 | 78.1 ± 1.6 |
| *T. plicata* | 15.1 ± 7.4 | 14.5 ± 7.7 | 0.0 ± 0.0 | 9.5 ± 1.9 | 68.9 ± 1.4 |
| *J. chinesis* | 4.1 ± 1.6 | 31.9 ± 5.5 | 49.0 ± 3.2 | 0.78 ± 0.12 | 74.0 ± 2.2 |
| *T. occidentalis* | 27.9 ± 4.5 | 28.0 ± 5.2 | 93.0 ± 0.7 | 29.2 ± 3.0 | 59.6 ± 2.7 |

**Table 3.b.** June/July 2018 (data are plotted in Figure 3.b).

| species | α-pinene | β-pinene | camphene | limonene | β-phellandrene |
|---|---|---|---|---|---|
| *P. menziesii* | 98.3 ± 0.4 | 99.9 ± 0.1 | 93.2 ± 1.1 | 71.3 ± 3.9 | 1.9 ± 0.4 |
| *P. ponderosa* | 99.5 ± 0.1 | 99.7 ± 0.2 | 85.6 ± 0.8 | 56.0 ± 12.6 | 1.9 ± 0.1 |
| *C. lawsoniana* | 1.9 ± 0.5 | 1.4 ± 0.3 | 0.0 ± 0.0 | 1.5 ± 0.2 | 81.0 ± 0.6 |
| *T. plicata* | 4.0 ± 2.0 | 15.0 ± 6.8 | 0.0 ± 0.0 | 6.5 ± 0.5 | 67.6 ± 0.9 |
| *J. chinesis* | 1.5 ± 0.5 | 12.2 ± 1.1 | 25.6 ± 2.9 | 0.42 ± 0.02 | 76.2 ± 2.5 |
| *T. occidentalis* | 24.1 ± 3.5 | 23.5 ± 8.4 | 93.2 ± 0.5 | 28.9 ± 0.8 | 57.1 ± 2.6 |

**Table 4.** Percent (–) enantiomer values ± 1 standard deviation (*s*) for five chiral monoterpenes in *Thuja occidentalis* in four nursery-purchased plants located at PSU and sampled once each (*N* = 4) in March 2018 and once each in June/July 2018. (Data are plotted in Figure 4.)

| time | *α*-pinene | *β*-pinene | camphene | limonene | *β*-phellandrene |
|---|---|---|---|---|---|
| 6 AM | 22.8 ± 2.3 | 33.3 ± 2.2 | 92.8 ± 0.4 | 27.6 ± 0.2 | 50.2 ± 0.1 |
| 1 PM | 24.8 ± 7.7 | 36.1 ± 6.1 | 92.7 ± 0.4 | 26.2 ± 0.9 | 51.9 ± 2.4 |
| 7 PM | 23.9 ± 1.8 | 32.4 ± 2.4 | 92.5 ± 0.6 | 27.7 ± 0.9 | 49.6 ± 1.3 |
| 9 PM | 24.2 ± 3.7 | 37.9 ± 6.1 | 92.6 ± 1.2 | 28.3 ± 1.7 | 47.5 ± 3.3 |

**Table 5.** Percent (–) enantiomer values ± 1 standard deviation (*s*) for five chiral monoterpenes in *Thuja occidentalis* in four nursery-purchased plants located at PSU and sampled once each (*N* = 4) in March 2018, and in residentially-planted samples found in a field trip to three suburban areas in Oregon (Seaside, *N* = 7 plants sampled once each; Hillsboro, *N* = 6 plants sampled once each; and Sandy, *N* = 7 plants sample once each). (Data are plotted in Figure 5.)

| location | *α*-pinene | *β*-pinene | camphene | limonene | *β*-phellandrene |
|---|---|---|---|---|---|
| PSU (purchased) | 27.9 ± 4.5 | 28.0 ± 5.2 | 93.0 ± 0.7 | 29.2 ± 3.0 | 59.6 ± 2.7 |
| Seaside (residential) | 28.4 ± 5.1 | 23.5 ± 9.3 | 94.4 ± 1.1 | 35.6 ± 2.9 | 62.5 ± 1.7 |
| Hillsboro (residential) | 24.1 ± 2.5 | 19.5 ± 2.6 | 92.2 ± 2.0 | 30.7 ± 1.9 | 62.7 ± 2.5 |
| Sandy (residential) | 22.1 ± 3.3 | 19.8 ± 3.8 | 94.1 ± 0.8 | 34.2 ± 3.0 | 58.3 ± 5.1 |