# Peer review of "Measurement of Enantiomer Ratios for Five Monoterpenes"

_Atmospheric Measurement Techniques, 2021_

## Author Comment (AC2)

Authors' Responses
June 1, 2022
**RC2 (Reviewer Comment 2)** on amt-2021-439
*"Measurement of Enantiomer Ratios for Five Monoterpenes From Six Conifer Species by Cartridge Tube-Based Passive Sampling Adsorption/Thermal Desorption (ps-ATD)"*
<Wang, Luo, Rosenstiel, Pankow>
* * *
**Comment 1.** *"The manuscript "Measurement of enantiomer ratios for five monoterpenes from six conifer species by cartridge tube-based passive sampling adsorption/thermal desorption (ps-ATD)" by Wang et al., presents a passive sampling desorption technique used to analyze specific VOCs from conifer/plant emissions. Specifically, five monoterpenes and their +/- enantiomers are analyzed. The manuscript is well presented and well organized. The main strength of this study is the presentation of a method that can be easy to use delivering sound results. The manuscript is overall worthy of publication after addressing a few minor comments."*
**Response 1. Accepted.** The authors thank the reviewer for this favorable comment.

**Comment 2.** *"What would increase the strength of this study is a comparison with existing methods for the detection of plant-emitted chiral biogenic VOCs. This topic is addressed in Table 1 but a discussion within the main text on how the existing methods compare to the proposed method will further highlight the importance of this study. It looks like other works in literature discuss the measurement of the same species measured here. How do they compare? What would make the suggested method better than what has been already used?"*
**Response 2. Accepted.** The authors agree that the text would benefit by addition of some elaboration of the material in Table 1. It had been our intention to save words by letting Table 1 speak for itself and logically carry the reader through to the conclusions that we had intended the reader reach, but we see now that this intent was perhaps demanding too much. Accordingly, in a revised manuscript, we can modify the text to lay this out in a logical manner, referring appropriately to the material in Table 1.

**Comment 3.** *"Line 45. I wouldn't just advertise the method precision as "excellent" unless an objective parameter is given so that the reader can see for himself/herself that the precision is indeed very good."*
**Response 3. Accepted.** The reviewer makes a good point that more elaboration is required in the paper on the method precision, and we are planning an enhanced discussion of the subject in a revised manuscript, particularly as regards how the method precision is affected both by the total abundance of an enantiomer pair and equally importantly by the percent of the (−) enantiomer. For the latter point, note here that when a (−) enantiomer is present at decreasing percentage of the +/− total, then the precision for that percentage will be degraded even as measuring the percentage of the + enantiomer becomes increasingly precise.

**Comment 4.** *"Line 74. Please give some example of which solvent are commonly used."*
**Response 4. No changes needed.** This information actually is already in Table 1 (hexane has been most common), but we can make this clear in a revised manuscript, per our response to Comment 2.

**Comment 5.**  "*Line 105. Is the altitude agl or asl?*"
**Response 5.  Accepted.**   The altitudes are "above sea level", and this will be made clear in a revised manuscript.

**Comment 6.**  "*Lines 106-108. I wonder if these last two sentences should be moved to the next section.*"
**Response 6.  Not Accepted.**   Each section needs its own explanation of how the samples were cut from the plants.

**Comment 7.**  "*Line 143. Please indicate which gas was used as carrier.*"
**Response 7.  Accepted.**   The carrier gas used was helium, and this information can easily be added in a revised manuscript.

**Comment 8.**  "Line 177. Should that "during" be eliminated?"
**Response 8.  Accepted.**    The first "during" in that line will be deleted in a revised manuscript.

**Comment 9.**  Line 189-190. Please give more details on why this assumption was considered excellent given that it was unverified.
**Response 9.  Accepted.**    It is well documented that gas molecules with the same molecular weight and same approximate volume have essentially the same gas phase diffusion coefficients in air.  This is demonstrated in Fuller's Equation, as discussed in some detail in https://acp.copernicus.org/articles/14/9233/2014/acp-14-9233-2014.pdf.  This can be explained per the reviewer's request in a revised manuscript.

---

## Author Response (AR1)

AMT-2021-439
Authors' Responses - June 21, 2022
*"Measurement of Enantiomer Ratios for Five Monoterpenes From Six Conifer Species by Cartridge Tube-Based Passive Sampling Adsorption/Thermal Desorption (ps-ATD)"*
<Wang, Luo, Rosenstiel, Pankow>

CHANGES IN REVISED MANUSCRIPT SHOWN IN GREEN HIGHLIGHT, NOT TRACK CHANGES
* * *
**RC1 (Reviewer Comments 1)**

**Comment 1.** *"Line 88: Please reference your active sampling ATD work."*
**Response.** Accepted.
**Authors' Changes.** See lines 84 and 285-286 in the revised manuscript.

**Comment 2.** *"Line 93: Don't need diffusive rates since identical is an assumption. This might be tested in future by measuring the rates."*
**Response.** Accepted.
**Authors' Changes.** None.

**Comment 3.** *"Line 122: Cut samples of plants instead of living plants sampled. Sampling the living plants in natural environment might have different results. This would be a good experiment to conduct."*
**Response.** Not accepted. This paper focuses on analytical method development, not biology.
**Authors' Changes.** None.

**Comment 4.** *"Can the time difference of when the cuts were made until analysis matter?"*
**Response.** Not accepted. This paper focuses on analytical method development, not biology.
**Authors' Changes.** None.

**Comment 5.** *"Figures 2-4- I didn't understand the ABC lettering system in the graph. I did understand capital and lowercase meant statistically insignificant for the monoterpenes, then the species, respectively but I didn't understand the a-f or A-E designations. Where these just random to demonstrate the relationships? Please provide abbreviations in the caption or as a footnote. Or consider including in the table summarizing the statistical significance."*
**Response.** Accepted.
**Authors' Changes.** Additional explanation has been added to the captions for Figures 2-5.

**Comment 6.** *"Also figures have error bars, please specify if these are 1 standard deviation or something else."*
**Response.** Accepted.
**Authors' Changes.** The requested explanation has been added to the captions for Figures 2-5.

**Comment 7.** *"Line 167 "was directed to split flow" may sound better."*
**Response.** Not Accepted.
**Authors' Changes.** None. The wording used is standard in the field.

**Comment 8.** *"Table 1: Last entry: Tenax spelled incorrectly (Tenas)"*

**Response.** Accepted.

**Authors' Changes.** The correction has been made.

**RC2 (Reviewer Comments 2)**

**Comment 1.** "*The manuscript "Measurement of enantiomer ratios for five monoterpenes from six conifer species by cartridge tube-based passive sampling adsorption/thermal desorption (ps-ATD)" by Wang et al., presents a passive sampling desorption technique used to analyze specific VOCs from conifer/plant emissions. Specifically, five monoterpenes and their +/- enantiomers are analyzed. The manuscript is well presented and well organized. The main strength of this study is the presentation of a method that can be easy to use delivering sound results. The manuscript is overall worthy of publication after addressing a few minor comments.*"

**Response. Accepted.** The authors thank the reviewer for this favorable comment.

**Authors' Changes.** None.

**Comment 2.** "*What would increase the strength of this study is a comparison with existing methods for the detection of plant-emitted chiral biogenic VOCs. This topic is addressed in Table 1 but a discussion within the main text on how the existing methods compare to the proposed method will further highlight the importance of this study. It looks like other works in literature discuss the measurement of the same species measured here. How do they compare? What would make the suggested method better than what has been already used?*"

**Response. Accepted.** The authors agree that the text would benefit by addition of some elaboration of the material in Table 1.

**Authors' Changes.** See line 77 in the revised manuscript.

**Comment 3.** "*Line 45. I wouldn't just advertise the method precision as "excellent" unless an objective parameter is given so that the reader can see for himself/herself that the precision is indeed very good.*"

**Response. Accepted.** The reviewer makes a good point that more elaboration is required in the paper on the method precision, and we are planning an enhanced discussion of the subject in a revised manuscript, particularly as regards how the method precision is affected both by the total abundance of an enantiomer pair and equally importantly by the percent of the (-) enantiomer. For the latter point, note here that when a (-) enantiomer is present at decreasing percentage of the +/- total, then the precision for that percentage will be degraded even as measuring the percentage of the + enantiomer becomes increasingly precise.

**Authors' Changes.** Section 3.5 has been totally rewritten, Figures 6 and 7 have been added, and Section 4 has been added. See lines 45, 244-258, and 412-413, in the revised manuscript.

**Comment 4.** "*Line 74. Please give some example of which solvent are commonly used.*"

**Response. Accepted.** This information actually is already in Table 1 (hexane has been most common), but we have made this clear in a revised manuscript, per our response to Comment 2.

**Authors' Change.** See line 77, Table 2, and lines 327-329 in the revised manuscript.

**Comment 5.** "*Line 105. Is the altitude agl or asl?*"

**Response. Accepted.** The altitudes are "above sea level", and this will be made clear in a revised manuscript.

**Authors' Changes.** See lines 115-116 and 127 in the revised manuscript.

**Comment 6.** "*Lines 106-108. I wonder if these last two sentences should be moved to the next section.*"
**Response. Not Accepted.** Each section needs its own explanation of how the samples were cut from the plants.
**Authors' Changes.** None.

**Comment 7.** "*Line 143. Please indicate which gas was used as carrier.*"
**Response. Accepted.** The carrier gas used was helium, and this information was added in the revised manuscript.
**Authors' Change.** See line 154 in the revised manuscript.

**Comment 8.** "*Line 177. Should that "during" be eliminated?*"
**Response. Accepted.** The first "during" in that line will be deleted in a revised manuscript.
**Authors' Change.** Correction made.

**Comment 9.** "*Line 189-190. Please give more details on why this assumption was considered excellent given that it was unverified.*"
**Response. Accepted.** It is well documented that gas molecules with the same molecular weight and same approximate volume have essentially the same gas phase diffusion coefficients in air. This is demonstrated in Fuller's Equation, as discussed in some detail in https://acp.copernicus.org/articles/14/9233/2014/acp-14-9233-2014.pdf.
**Authors' Changes.** See lines 97, 197, and 324-326 in the revised manuscript.